# Barriers and Factors Affecting the E-Commerce Sustainability of Thai Micro-, Small- and Medium-Sized Enterprises (MSMEs)

Yot Amornkitvikai [1] , Siew Yean Tham [2] , Charles Harvie [3] and Wonlop Writthym Buachoom [4],*

1   College of Population Studies, Chulalongkorn University, Bangkok 10330, Thailand; yot.a@chula.ac.th
2   Regional Economic Studies, ISEAS—Yusof Ishak Institute, Heng Mui Keng Terrace, Singapore 119614, Singapore; siew_yean@iseas.edu.sg
3   Economics Discipline, Faculty of Business and Law, University of Wollongong, Wollongong, NSW 2522, Australia; charvie@uow.edu.au
4   KMITL Business School, King Mongkut's Institute of Technology Ladkrabang, Bangkok 10520, Thailand
*   Correspondence: wonlop.bu@kmitl.ac.th

**Abstract:** It is anticipated that e-commerce will contribute to achieving the 17th Sustainable Development Goal, which seeks to improve implementation mechanisms and revitalize global partnerships for sustainable development. However, MSMEs still face a digital gap compared to large enterprises, which affects their e-commerce sustainability. The study's objective is to examine the factors and barriers affecting the e-commerce sustainability of Thai micro-, small- and medium-sized enterprises (MSMEs) based on a survey of retail and food and beverage (F&B) service MSMEs in metropolitan Bangkok. Estimations confirm the significance of the TOE framework for Thai MSMEs. Internal e-commerce tools (i.e., smartphones and websites) and external e-commerce platforms (i.e., social media, e-marketplaces, and food delivery platforms) can enhance e-commerce sustainability. However, the age of firms and owners (CEOs) affects e-commerce sustainability negatively. Exports, B2B e-commerce, and e-commerce experience can promote the e-commerce sustainability of Thai MSMEs. However, they perceive that many consumers are still not literate in using e-commerce. In addition, Thailand still has insufficient security to prevent hacking and malware. Therefore, Thai entrepreneurs' e-commerce literacy is insufficient to enhance their e-commerce sustainability. On the other hand, sustainable e-commerce can increase customer satisfaction, loyalty, and trust through customer support, leading to more long-term online shopping. Hence, this study focuses on e-commerce sustainability-based economic dimensions, as measured by the percentage of e-commerce sales to total sales (e-commerce utilization/intensity).

**Keywords:** e-commerce; micro-, small- and medium-sized enterprises (MSMEs); sustainability; retail; food and beverage; Thailand

## 1. Introduction

The transformation of global digitalization from the development of high-speed Internet (4G/5G), smartphones, facilitation of online payments, shifts in consumer behavior, and services sector liberalization has significantly improved cross-border e-commerce, resulting in greater levels of e-commerce transactions [1]. Similarly, the exponential growth of Internet users and new social networking platforms (i.e., Facebook, Twitter, Instagram, blogs, and WhatsApp) has moved the traditionally offline market to a modern customer market structure [2]. The rapid change in the global e-commerce ecosystem has increased to more than 1.4 billion online consumers [3]. Global retailers have responded by adopting e-commerce businesses for their tech-savvy and time-conscious customers [4]. Global e-commerce sales reached USD 25,648 billion, accounting for 30 percent of global gross domestic product (GDP). In particular, B2B e-commerce sales accounted for the majority of all e-commerce sales, totaling USD 21,258 billion, or 24.87 percent of the global GDP [3]. Globally, e-commerce sales in the retail industry reached 3914 billion U.S. dollars

in 2020. Due to China's dominance in e-commerce, Asia-Pacific accounted for 62.6 percent of global retail e-commerce sales, followed by North America (19.1 percent), Western Europe (12.7 percent), Central and Eastern Europe (2.4 percent), Latin America (2.1 percent), and the Middle East and Africa (1.1 percent) [5,6].

Notably, e-commerce is anticipated to contribute to the achievement of the 17th Sustainable Development Goal, which seeks to improve the implementation mechanisms and energize international partnerships for sustainable development. By 2020, e-commerce can increase exports from developing countries and double the proportion of exports from the world's least developed countries [7,8]. Following the 5th Sustainable Development Goal, e-commerce can also be used to promote entrepreneurship, including empowering female entrepreneurs, which can reduce gender inequality and promote the rights of all women in society. E-commerce has increased globally since the emergence of the COVID-19 pandemic due to the enforcement of social distancing, lock-downs, and other measures used to combat the global health crisis [9]. The pandemic has changed global consumer behavior towards online channels, permanently changing the e-commerce market landscape [10].

E-commerce plays a critical role in driving economic growth in Thailand. E-commerce is one of the fastest-growing businesses in recent years [11]. It attracts investments that support the development of the Thai digital infrastructure. It also helps Thai businesses to become more innovative, creates more high-income employment throughout the supply chains, and encourages more skilled workers [12]. In addition, it further stimulates household consumption, reduces the digital divide between rural and urban areas, and improves the country's asymmetry in market information.

As part of Thailand's Industry 4.0 Policies, the government has lent support to electronic activities by promoting e-commerce platforms and social media applications [13]. In addition, digital infrastructure, such as the nationwide broadband network, has been promoted as one of the critical pillars in Thailand's Industry 4.0 policies, aiming to bridge the digital divide and promote the country's modern economic development via e-commerce [14].

Moreover, e-commerce can help rapidly connect MSMEs in developing countries to global markets [15]. However, Thai e-commerce entrepreneurs still rely mainly on the domestic market. In 2018, 91.3 percent of total online sales were domestic, while the remaining 8.7 percent originated from the international market [16]. More importantly, Thai MSMEs' e-commerce values increased significantly from USD 14,961.4 million in 2017 to USD 39,039.4 USD million in 2018 as more Thai MSMEs and customers veered towards increasing online purchases and sales. Nonetheless, Thai MSMEs' e-commerce values are still lower than that of large enterprises, accounting for 35.2 percent of total e-commerce values in 2019, as indicated in Figure 1.

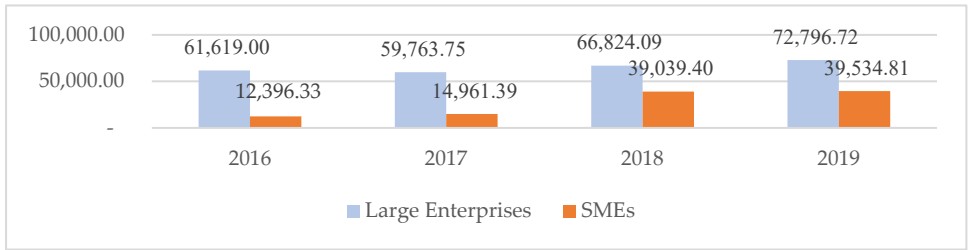

**Figure 1.** Thailand's value of e-commerce (2016–2019). Unit: USD million. Source: Electronic Transactions Development Agency (ETDA) [16].

Thailand's e-commerce sector (including large enterprises and SMEs) is valued at USD 112,331.53 million in 2019, increasing 6.1 percent compared to 2018, accounting for 31.2 percent of GDP. Within e-commerce, business-to-business (B2B) e-commerce has the largest share in 2019, accounting for 47.5 percent of the total e-commerce value, followed by business-to-consumer (B2C) e-commerce (37.2 percent) and business-to-government (B2G) e-commerce (47.2 percent) [16]. E-commerce transactions are mainly concentrated in

the domestic market, accounting for 91.3 percent of total online products and services in 2019 [16].

Nonetheless, Thai MSMEs still face several barriers to e-commerce adoption since they still utilize outdated tools and do not have access to the requisite technology for adopting e-commerce. In terms of e-business usage levels, Thai MSMEs are far behind in adopting e-business technology. Therefore, there is a so-called digital gap between large and small firms in adopting new technology [17–19], resulting in lower levels of e-commerce value compared to large enterprises, as shown in Figure 1. In the e-commerce industry, MSMEs still cannot access many customers and suppliers relative to large firms in the same sector which are operating internationally. They also lack marketing knowledge and experience in adopting e-commerce [20]. These constraints can inhibit Thai MSMEs from adopting e-commerce to enhance their revenues.

This paper aims to investigate empirically the factors that influence the e-commerce sustainability of Thai MSMEs in food and beverage and retail services. A few earlier studies have investigated e-commerce adoption in Thailand. Changchit et al. [21] used a survey of customers to examine the factors that affect shopping online in Thailand, while the other studies used survey data collected from Thai MSMEs. Sutanonpaiboon and Pearson [22] investigated the factors affecting e-commerce adoption by 400 Thai MSMEs. Their results suggest that Thai MSMEs lack organizational readiness. Many Thai MSMEs lack preparedness to adopt e-commerce due to technological, cultural, financial, and logistical difficulties. Lertwongsatien and Wongpinunwatana [23] used ANOVA analysis to examine factors influencing e-commerce adoption by Thai MSMEs. They separated their sample of 452 respondents into three categories of e-commerce readiness: adopters, prospectors, and laggards.

Environmental, social, and economic dimensions enable practical and sustainable e-commerce conformity. Economic sustainability is defined as how business owners should economize available resources to maximize profits and ensure adequate revenue for stakeholders. It implies that companies should focus on short-term and long-term goals by ensuring that customers obtain value for their money and are impressed with their purchases [24].

Sustainable e-commerce can be achieved by optimizing asset utilization, controlling factory production and administration costs, and responding nimbly to changing consumer demands, thereby increasing customer satisfaction, loyalty, and trust through customer support. Customers may abandon online shopping and return to brick-and-mortar stores if e-commerce firms cannot sustain themselves and meet their needs [24]. However, while previous empirical studies examined e-commerce adoption by Thai MSMEs, they did not focus on e-commerce sustainability-based economic dimensions, as measured by the percentage of e-commerce sales to total sales (e-commerce utilization/intensity).

Unlike those conducted in other countries, this study attempts to quantify the barriers and factors affecting Thai MSMEs' e-commerce sustainability (intensity), focusing on sustainability's economic dimensions. The findings of this study can be applied to other developing nations, as Thailand is a good representative of Asian countries. However, Thai MSMEs still face several barriers to e-commerce adoption due to their continued use of obsolete tools and lack of access to the technology required for e-commerce adoption. In e-business technology adoption, Thai MSMEs lag significantly behind international peers.

This study aims to fill this research gap by adopting the TOE framework to investigate factors affecting Thai MSMEs' e-commerce sustainability in the Bangkok Metropolitan Region in the food and beverage and retail services. Previous studies in e-commerce adoption focused on binary outcomes, that is, e-commerce adoption and non-adoption. The disadvantage of such a binary outcome is that it does not adequately inform the technology issue [25–27]. In addition, the binary outcome cannot capture the levels of e-commerce sustainability, leading to an imprecise analysis in previous studies on e-commerce adoption. The model in this study, on the other hand, captures the whole range of e-commerce sustainability as it measures e-commerce sustainability as the percentage of e-commerce

sales to total sales. Due to MSMEs' highly centralized decision-making structures, this study also introduces the owner's (CEO) characteristics as one of the critical organizational factors to ascertain their impact on e-commerce sustainability (intensity).

The outline for this paper is as follows. Section 2 reviews the literature and constructs the hypotheses for testing as well as the conceptual framework. Section 3 discusses the data and methodology adopted in this paper, while the empirical results and discussions are revealed in Sections 4 and 5. Theoretical and practical implications are given in Section 6. Section 7 summarizes the main findings in conclusion. Limitations and future studies are discussed in the last section.

## 2. Literature Review and Hypotheses Development

This section reviews the theories adopted in empirical studies of technology adoption, as indicated in Section 2.1, and previous empirical studies based on the technology–organization–environment (TOE) framework, as shown in Section 2.2.

### 2.1. A Review of Theories

This section examines the theories utilized in technology adoption studies. Three major theories have been applied to the study of technology adoption: (i) the technology adoption model (TAM), which refers to users' acceptance of adopting technology based on their perception of its usefulness and ease of use, as proposed by Davis et al. [28]; (ii) the diffusion of innovation theory (DOI), which suggests how innovation is communicated through communications by the population in a social system, as suggested by Rogers [29]; (iii) the technology-organization-environment (TOE) framework that refers to the adoption decision of technological innovation based on technological, organizational, and external environmental con-texts, as proposed by Tornatzky and Fleischer Tornatzky and Fleischer [30]. The TOE framework by Tornatzky and Fleischer [30] has been effectively applied to the literature on e-commerce adoption, which focuses more on firm-level analysis [25,31–36].

The technological context in the TOE framework refers to a company's existing and new technology. In contrast, the organizational context encompasses descriptive measurements such as the organization's scope, size, age, and managerial beliefs. Environmental context refers to external factors, such as government incentives and regulations, influencing business operations [25]. For MSMEs, owners can make centralized decisions solely by themselves to lead their firms. In response to the highly centralized decision-making structures of MSMEs, Thong and Yap [35] and Thong [34] introduced CEO characteristics as a fourth dimension, which is subsequently used for studies focusing on small businesses such as Safari et al. [37] and Imr [38]. This study also introduces the CEO characteristics, but they are included as a part of the organizational factors as in Al-Qirim [39]. More importantly, the TOE framework is more oriented toward firm-level analysis. The TOE framework has been adopted in empirical studies, as reviewed in the following section.

### 2.2. Factors Influencing E-Commerce Adoption Based on the TOE Framework

This section reviews the literature related to empirical studies which examined the technological, organizational, and environmental factors based on the TOE framework. Section 2.2.1 discusses technological factors (internal e-commerce tools), Section 2.2.2 discusses organizational factors, consisting of (i) the owner's (CEO) characteristics, (ii) exports and B2B e-commerce, and (iii) firms' characteristics. Section 2.2.3 then discusses technological factors (external e-commerce platforms). Finally, the study's hypotheses are provided at the end of each sub-section.

2.2.1. Technological Factors (Internal E-Commerce Tools) and Hypothesis Development

Technological factors can include computers, smartphones, and websites, as indicated by the firms' internal e-commerce tools. Moreover, based on empirical studies in the literature, these technological tools can significantly affect e-commerce adoption [40–44].

For firms' e-commerce tools, Walker, Saffu, and Mazurek [41] found that PC networks and internet providers for e-commerce adopters are statistically different from non-adopters. Smartphones have given niche markets for e-commerce users, especially mobile commerce, due to the high rates of smartphone penetration around the globe [45]. Smartphones are deemed essential for conducting online business [46]. Apergis [46] also indicated that owners in family businesses tend to adopt tablets and smartphones to operate their businesses, but younger entrepreneurs prefer to use smartphones. Rahayu and Day [47] also revealed that technological readiness significantly and positively influences e-commerce adoption by Indonesian MSMEs.

Salehi et al. [48] further suggested that websites with clear information about firms' products and services can attract customers, leading to the success of their e-commerce sales. Websites can be used as an effective e-commerce tool for marketing and selling goods and services. Furthermore, the quality of websites can enhance customers' satisfaction and loyalty [49]. Walker, Saffu and Mazurek [41] found that website usage is statistically different between e-commerce adopters and non-adopters. Shaltoni [50] also highlighted that websites are essential for all enterprises since websites can provide superior search engine marketing. In addition, firms can comprehensively show their product content using websites, resulting in better branding [50,51]. Nevertheless, decision makers may not prefer websites, especially for small firms, as it is relatively more time-consuming and expensive to maintain their dedicated websites [50]. However, in Thailand, many firms have adopted websites to sell online via the Internet [52].

As discussed earlier, firms can achieve sustainable e-commerce economic dimensions by optimizing asset utilization, controlling factory production and administration costs, responding nimbly to changing consumer demands, and increasing customer satisfaction, loyalty, and trust through customer support [24]. However, improving the economic aspect of e-commerce sustainability results from more online shopping customers. [24]. This will lead to more e-commerce utilization/intensity of Thai MSMEs. As a result, the first hypothesis is:

**Hypothesis 1 (H1).** *The firm's e-commerce tools positively affect the e-commerce sustainability (intensity) of Thai MSMEs.*

### 2.2.2. Organizational Factors and Hypothesis Development

- Owner's (CEO) characteristics

Some empirical studies have included several characteristics of the owners or the CEOs as variables that can affect e-commerce adoption. These are (i) the owner's (CEO) age, (ii) owner's (CEO) gender, (iii) owner's (CEO) education, and (iv) owner's (CEO) IT skills. For instance, Scupola and Dwivedi [33] pointed out that top management support and CEO factors are critical factors that affect e-commerce adoption and implementation in the case of Danish and Australian MSMEs. Walker, Saffu, and Mazurek [41] revealed that the owner's gender, experience, and education of e-commerce adopters are statistically different from non-adopters for Slovakian MSMEs.

Information technology (IT) skills can be an important determinant for e-commerce adoption. Rahayu and Day (2015) found that owners' innovation, IT ability, and IT experience are critical factors for Indonesian MSMEs' e-commerce adoption. Ghobakhloo et al. [53] also found that a CEO's innovativeness has a significant and positive impact on e-commerce adoption by managers or owners of manufacturing MSMEs in Iran. Muathe and Muraguri-Makau [54] revealed that CEOs with ICT knowledge are likely to adopt e-commerce. Lip-Sam and Hock-Eam [55] also found that owners or CEOs who use computers frequently are likely to adopt e-commerce. The Davis and Matteis [56] findings recommend firms to use managers with hybrid business experience and technology skills to operate e-commerce transactions. Hence, owners need a certain level of IT and e-business skills to operate their e-commerce businesses successfully.

The owner's (CEO) age can significantly affect IT and e-commerce adoption by MSMEs. In the study by Nair et al. [57], the owner's age is significant for IT preparedness and implementation. Muathe and Muraguri-Makau [54] also found that a CEO's age is significantly and negatively related to e-commerce adoption by MSMEs in Kenya. Similarly, Chuang et al. [58] revealed that a CEOs' age, on average, is significantly and negatively related to the IT adoption by MSMEs in the US. These results imply that older owners (CEOs) are less likely to adopt e-commerce compared with younger ones, due to their limitations in adopting new technologies for their business. By contrast, Awa et al. [59] found that the average age of CEOs positively affects IT adoption by MSMEs in Nigeria.

Focusing on the effects of CEO's education on IT and e-commerce adoption, Lip-Sam and Hock-Eam [55] also used the TOE framework to investigate the factors affecting e-commerce adoption by MSMEs in Malaysia. They found that owners or CEOs with more experience and tertiary education are more likely to adopt e-commerce. Chuang, Dwivedi, Nakatani and Zhou [58] found that education is significantly and positively associated with IT adoption by MSMEs in the US. On the other hand, Mahliza [42] found that owners' knowledge and skills are not significantly related to e-commerce adoption among Indonesian MSMEs in the Jakarta region. Similarly, Awa, Jones, Eze, Urieto and Inyang [59] also found that education is not correlated with IT adoption by MSMEs in Nigeria. Mixed results are found from previous studies on the impact of CEO's gender on their firms' IT adoption. For instance, Riding [60] explored the impact of CEO's gender on small firms' information technology (IT) adoption and found that female CEOs are less likely than men to adopt IT in the small firm context. Similarly, Awa, Jones, Eze, Urieto and Inyang [59] investigated the effects of CEO's characteristics on MSMEs' information technology (IT) adoption. They revealed that male CEOs are likely to be more inclined to adopt IT than their female counterparts. Yusgiantoro et al. [61] also concluded that male owners are more likely to adopt e-commerce than female MSME owners in Indonesia. Uzoka et al. [62] also found that female CEOs are likely to affect e-commerce adoption negatively in developing countries.

The owner's IT skills mean owners (CEOs) with a degree in science, information technology (IT), and engineering represent the owner's IT skills. In addition, the owner's education refers to owners (CEOs) who obtained at least a bachelor's degree. This study hypothesizes the following second hypothesis to examine whether an owner's (CEO) characteristics (i.e., the owner's IT skills, owner's education, owner's gender, and owner's age) are significantly associated with their respective firms' e-commerce sustainability (intensity):

**Hypothesis 2.1 (H2.1).** *The owner's IT skills positively affect the e-commerce sustainability (intensity) of Thai MSMEs.*

**Hypothesis 2.2 (H2.2).** *The owner's education positively affects the e-commerce sustainability (intensity) of Thai MSMEs.*

**Hypothesis 2.3 (H2.3).** *The owner's gender (female owner) negatively affects the e-commerce sustainability (intensity) of Thai MSMEs.*

**Hypothesis 2.4 (H2.4).** *The owner's age negatively affects the e-commerce sustainability (intensity) of Thai MSMEs.*

- Exports and B2B e-commerce

Exporting as part of globalization and international engagement is correlated with e-commerce transactions [63,64]. Gautam [65] found a positive relationship between e-commerce adoption and export sustainability for Indian manufacturing firms, as e-commerce can minimize export costs. Kraemer, Gibbs, and Dedrick [64] also found a positive association between globalization, international exposure, and e-commerce adoption. Likewise, there is a positive association between the volume of international trade

and e-commerce in Terzi [63]. In addition, exporting firms can benefit from their learning by exporting experience due to knowledge transfers from their foreign counterparts to these firms [66]. For Thailand, Ueasangkomsate [67] also found that Thai MSME exporters can realize and reap the benefits of e-commerce adoption. However, since few have fully utilized their e-commerce capabilities, e-commerce adoption has no significant impact on export sustainability in his study.

Moreover, B2B e-commerce can promote domestic and cross-border e-commerce, thereby increasing a firm's e-commerce transactions. Firms that adopt B2B e-commerce can improve business opportunities and efficiency via communications with their distributors and suppliers in the supply chain [68]. Empirical studies also suggest a positive association between B2B and cross-border e-commerce and international trade [69–71]. For instance, Wang, Wang, and Lee [69] found that cross-border e-commerce positively affects foreign trade in China. In addition, cross-border e-commerce can reduce Chinese firms' transaction costs and avoid paying tariffs for small parcels. UNCTAD [70] also showed that more B2B cross-border e-commerce could significantly increase global trade in goods and services. Askar [71] further pointed out that cross-border e-commerce is expected to replace the traditional market and become a significant contributor to global trade. Cross-border e-commerce can help firms achieve global connections through the Internet with free, open, and universal trade platforms. In addition, B2B business positively impacts cross-border e-commerce in China, thereby increasing exports. For Thailand, e-commerce is largely driven by B2B e-commerce rather than B2C e-commerce and B2G e-commerce transactions. In addition, B2B e-commerce also plays a significant role in supply chains in Thailand [72,73]. Iyer et al. [74] found a significant and positive association between B2B supply chain integration and US firms' sales or market performance. Finally, e-commerce can reduce transaction costs (i.e., labor and administrative costs) dealing with trading partners and enhance firms' production and delivery efficiency. Given the above, this study suggests the following third and fourth hypotheses:

**Hypothesis 3 (H3).** *E-commerce exports positively affect the e-commerce sustainability (intensity) of Thai MSMEs.*

**Hypothesis 4 (H4).** *Business to business (B2B) positively affects the e-commerce sustainability (intensity) of Thai MSMEs.*

- Firm characteristics

Firm size and firm age are critical firm-specific factors that can affect the e-commerce sustainability of Thai MSMEs. Larger firms are likely to benefit from a firm's economies of scale and scope, and older firms also benefit from their learning-by-doing experience. However, mixed results are obtained from empirical studies. For example, in their ten-country study of 2139 firms, Kraemer, Gibbs, and Dedrick [64] found that firm size is significantly and positively associated with the scope of e-commerce. Walker, Saffu, and Mazurek [41] found that the firm size of e-commerce adopters is larger than non-adopters. Uzoka, Seleka, and Khengere [62] also found that firm size is significantly and positively associated with e-commerce adoption in developing countries. However, Hong and Zhu [75] found that firm size is negatively associated with the e-commerce adoption of 1035 US and Canadian firms in their study. For Indonesia, Rahayu and Day [47] found that firm size does not significantly impact MSMEs' e-commerce adoption. For Thailand, Lertwongsatien and Wongpinunwatana [23] and Brown and Kaewkitipong [19] found that firm size is significantly and positively related to e-commerce adoption and e-business use of Thai MSMEs. Despite inconclusive results in the literature, the ETDA [11] pointed out that smaller Thai firms lack the awareness and capability to use ICT for their businesses, implying that size can negatively affect e-commerce adoption.

A firm's age for e-commerce adopters is statistically different from non-adopters in Walker, Saffu and Mazurek [41]'s study. Firm age can hinder organizational readiness, as

revealed by Nair, Chellasamy, and Singh [57]. Therefore, older MSMEs tend to face more significant e-commerce adoption difficulties than younger ones due to their limitation in adopting IT. Hence, older MSMEs will likely prefer brick-and-mortar (offline) sales over online sales as they may not be familiar with the requisite new technologies.

Firms with more experience in adopting e-commerce will gain e-commerce capabilities than those with less or no e-commerce experience due to accumulative know-how [76]. Furthermore, firms with more e-commerce experience will have more time and resources to reap the benefits from their e-commerce transactions. Hence, their e-commerce experience exerts a positive influence on e-commerce. Gregory et al. [77] pointed out that the e-commerce export experience significantly increases a firm's production degree, promotes adoption, enhances communication and distribution efficiencies, and improves price competitiveness for export ventures in Austria. Nonetheless, Ramanathan, Ramanathan, and Hsiao [76] found contradictory evidence as Taiwanese MSMEs in their study with more e-commerce experience do not significantly affect e-commerce performance compared to less experienced MSMEs. However, e-commerce experience is assumed to enhance the export sustainability of Thai MSMEs due to their cumulative learning-by-doing experience. To examine a firm's characteristics influencing the e-commerce sustainability, this study hypothesizes the following:

**Hypothesis 5.1 (H5.1).** *Firm size positively affects the e-commerce sustainability (intensity) of Thai MSMEs.*

**Hypothesis 5.2 (H5.2).** *Firm age negatively affects the e-commerce sustainability (intensity) of Thai MSMEs.*

**Hypothesis 5.3 (H5.3).** *Firm e-commerce experience positively affects the e-commerce sustainability (intensity) of Thai MSMEs.*

2.2.3. Environmental Factors (External E-Commerce Platforms) and Hypothesis Development

External e-commerce platforms are provided by social networking and online platform providers. Small firms are likely to prefer social media via Facebook, Line, and Instagram, since it is relatively easy to manage without any costs, especially for their e-commerce. Furthermore, social networks can increase firms' network sustainability based on environmental factors. Firms can expand their customer base and establish e-commerce platforms via online social networks [78]. Mahliza [42] and Lea, Yu, Maguluru, and Nichols [43] found that social media is necessary for e-commerce adoption. Likewise, Derham, Cragg, and Morrish [44] also found that social media is a valuable e-commerce platform for MSMEs. Social media is easily accessible with low barriers to entry; it requires low IT skills for access and can minimize the firms' costs. Additionally, Facebook usage positively influences MSMEs' performance. It can reduce the firms' marketing and customer services costs, build customer relations, and enhance the accessibility of information [79]. However, social media pages cannot guarantee that firms will succeed when there is a lack of a strategic perspective [80]. As Tarsakoo and Charoensukmongkol [81] suggest, out of the five social media marketing capabilities, social media's product development and marketing implementation capability can significantly improve customer relationships and the financial performance of Thai companies.

According to Pasadilla et al. [82], MSMEs can access larger markets and obtain necessary information for exports via e-marketplace, particularly cross-border trade. Therefore, e-marketplaces can be an external e-commerce platform that can help firms increase their sales to the e-market and increase market transparency [83]. Moreover, e-marketplaces can monitor and protect the theft across the supply chain, thereby enhancing supply chain efficiency [83]. Therefore, online food-delivery platforms have become an essential vehicle for Thai MSMEs for food and beverage services. This is because they provide an accessible marketplace and increase sales [84,85]. Hence, this study suggests the following hypothesis:

**Hypothesis 6 (H6).** *External e-commerce platforms positively affect the e-commerce sustainability (intensity) of Thai MSMEs.*

### 2.3. Technology–Organization–Environment (TOE) Barriers and Hypothesis Development

Empirical studies on the identification of different types of barriers for e-commerce adoption. Studies using the TOE and/or its extended framework have different findings despite using a common framework, as the items tested in each dimension differ from one study to another [25,33,47,53,86,87]. Overall, in terms of the framework, empirical studies conducted in Southeast Asia indicate the importance of the organizational and external environment context in e-commerce adoption [87], although the specific items within each dimension, which are significant, varied from one study to another.

In studies using the non-TOE framework, Saif-Ur-Rehman and Alam [31]'s model classified e-commerce barriers into five categories, such as (i) organizational, (ii) financial, (iii) technical, (iv) legal and regulatory, and (v) behavioral barriers. These e-commerce barriers are found to influence Malaysian MSMEs' e-commerce adoption using survey data. Specifically, legal and regulatory hurdles were the most critical barriers, followed by technical, financial, behavioral, and organizational obstacles. Lawrence and Tar [32] also used a non-TOE framework based on four key barriers, namely, (i) specific infrastructure barriers, (ii) socio-cultural barriers, (iii) socioeconomic barriers, and (iv) political and government barriers. Using macro data, their results show that the unavailability of adequate necessary infrastructure, socioeconomic barriers, and the lack of national ICT strategies have caused significant obstacles to e-commerce adoption and growth of e-commerce in developing countries. Kshetri [88] also used a non-TOE framework for a case study of a Nepalese online provider. There are three main types of e-commerce barriers: (i) economic barriers, such as unreliable and expensive power, lack of ICT infrastructure and use, limited use of credit cards, low purchasing power, and weak financial systems; (ii) socio-political barriers, such as weak legal and regulatory frameworks, cultural preferences for face-to-face interaction, and a society's reliance on cash; (iii) cognitive barriers, which stem from a lack of ICT literacy, awareness, and e-commerce-related knowledge among both consumers and businesses. Abualrob and Kang [89] classified the barriers into external and internal barriers. Governmental instability, occupational restrictions, and logistical obstacles are external barriers. Perceived losses, perceived uncertainty, and perceived complexity are internal barriers. Their research indicates that occupation restrictions and political factors are the primary obstacles that could prevent Palestinian business owners from adopting e-commerce. MacGregor and Vrazalic [90] generally grouped several e-commerce adoption barriers into two main categories for small businesses in Sweden and Australia. Firstly, "too difficult" barriers include several items in the questionnaire, such as (i) lack of technical knowledge in the organization, (ii) e-commerce being too complex to implement, (iii) required financial investments are too high, (iv) lack of time to implement e-commerce, and (v) difficulty selecting among various e-commerce options. Second, "unsuitable" barriers refer to the group of items in the questionnaire, such as (i) not suited to products and services, (ii) not suited to a method of conducting business, (iii) not suited to clients' method of conducting business, and (iv) no benefits from e-commerce. They discovered that barriers to e-commerce adoption could be attributed to two factors: either e-commerce is too challenging to implement or unsuitable for the business.

**Hypothesis 7 (H7).** *Technological, organizational, and environmental (TOE) barriers can hinder the e-commerce sustainability (intensity) of Thai MSMEs.*

According to the literature review and hypothesis development, the conceptual framework can be summarized in Figures 2 and 3, based on the TOE framework.

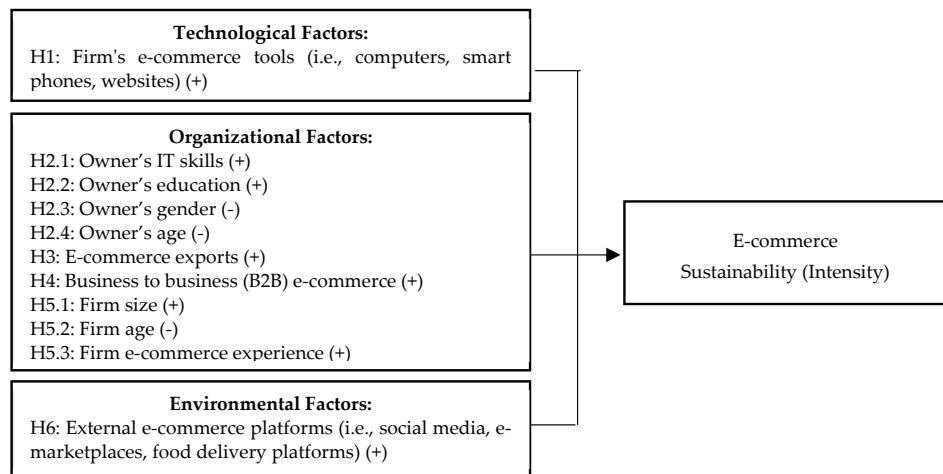

**Figure 2.** A conceptual framework based on the TOE framework: Factors affecting e-commerce inability Source: Authors. Note: The expected sign for each hypothesis is shown in the parathesis.

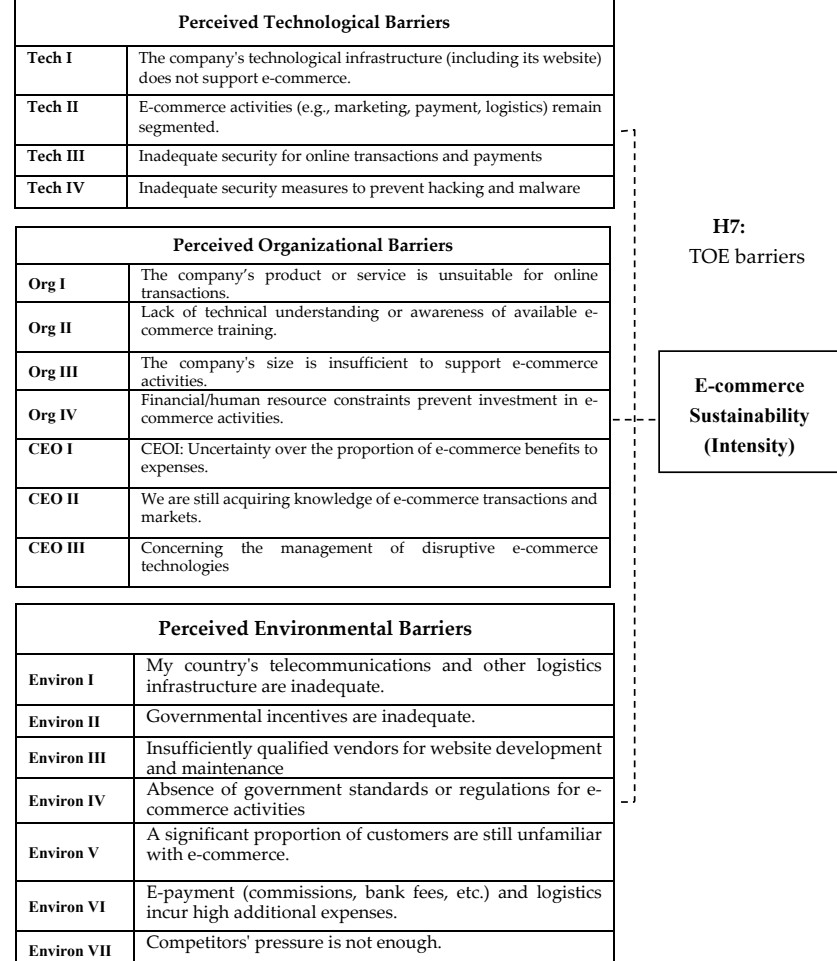

**Figure 3.** A conceptual framework based on the TOE framework: Identifying key barriers hindering the e-commerce sustainability of Thai MSMEs. Source: Authors.

## 3. Data and Methodology

### 3.1. Data Collection

Based on 2019 Thailand's criteria for MSMEs in service and merchandising, any firm that hires five or fewer employees is classified as a micro-enterprise. In comparison, a

firm hiring 6 to 30 employees is classified as a small enterprise, while a firm hiring 31 to 100 employees is classified as a medium enterprise [91].

This study collected the primary data of 307 MSMEs in the retail and food and beverage sectors (this study focuses only on retail and F&B services sectors, due to increasing e-commerce transactions and the importance of these sectors) by adopting the 2019 definition of MSMEs, as shown in Table 1. In addition, Thai MSMEs in the retail sector, excluding food and beverage services (i.e., clothes, accessories, and home decorations), were selected as the respondents. However, this study collected the data from those in the F&B services sector (i.e., those selling F&B in bars, restaurants, canteens, food courts, cafeterias, and other food-based hospitality businesses). Levine et al. [92] suggested that a minimum of 271 Thai MSMEs is needed to satisfy a 10 percent confidence level and a 5 percent margin of error.

**Table 1.** 2019 Definition of MSMEs.

| Sector | Micro and Small Enterprise | | | | Medium Enterprise | |
| | Micro Enterprise | | Small Enterprise | | | |
| | Annual Income (Million Baht) | Employment (Person) | Annual Income (Million Baht) | Employment (Person) | Annual Income (Million Baht) | Employment (Person) |
| --- | --- | --- | --- | --- | --- | --- |
| Manufacturing | $\leq 1.8$ | $\leq 5$ | $\leq 100$ | $\leq 50$ | $\leq 1.8$ | $\leq 5$ |
| Service and Merchandising | $\leq 1.8$ | $\leq 5$ | $\leq 50$ | $\leq 30$ | $\leq 300$ | $\leq 100$ |

Source: The Office of SMEs Promotion (OSMEP) [82]. Note: The new definition of MSMEs is subject to Ministerial Regulations on Designation of the Characteristics of SME Promotion Act BE 2562 (2019).

The survey questionnaire was initially developed based on conceptual framework and literature review (see Appendix A). Later, it was confirmed through a stakeholder meeting at which government and business representatives involved in e-commerce in Singapore were invited to provide constructive criticism on the survey questionnaire. The stakeholder meeting was held in Singapore at the ISEAS—Yusof Ishak Institute. Finally, the pilot survey was conducted in Thailand with ten respondents, to allow for expert review and ensure that each respondent correctly understood each question.

Appendix A outlines the structure of the survey questionnaire used in this study. Based on the TOE framework, this study used Appendix A.1's questions to identify significant factors affecting the sustainability of e-commerce (intensity) that impact its intensity (see Appendix A.1). In addition, this research employed sub-barrier questions in Appendix A.2 to identify and assess the most significant obstacles to the sustainability of e-commerce (intensity). The purposive sampling technique or non-probability sampling is adopted in this study. It does not require random samples [93,94]. More specifically, expert sampling as a part of purposive sampling is used in this study, since only CEOs, owners, or senior managers are the respondents who answered the questionnaire.

Consequently, between November 2018 and February 2019, face-to-face interviews and purposive sampling were used to collect data from 307 MSMEs operating in the retail and food and beverage service industries within the Bangkok Metropolitan Region. As a result, there are 177 retail respondents (57.7 percent of the total) and 130 food and beverage (F&B) services respondents (42.4 percent of total respondents).

### 3.2. Methodology for Econometric Model

This study employs the quantitative technique, including the Tobit regression model and the One-Sample Wilcoxon signed-rank test. (In the quantitative technique, unlike qualitative analysis, a larger, randomly selected sample increases the likelihood that quantitative findings can be generalized to the entire population or a subpopulation of the population [95]. In addition, the findings of quantitative studies are based on the sampling distribution theory. Specifically, one can generalize the results of a study to the extent that the sample is representative of the population from which it was drawn. Consequently, the

findings of a quantitative study can (theoretically) be applied to individuals with similar characteristics to those in the original study. Hence, replication is a cornerstone of quantitative research since it identifies fraud and invalid findings. If a study cannot be replicated, it is deemed an anomaly, a fluke, or to have methodological flaws. Even when quantitative studies are used to replicate findings, the results are never identical. Random and chance variables cause results to fluctuate and vary [96]. If quantitative researchers demonstrate a 95-out-of-100 probability of repeating results, this is generally regarded as a successful replication in the social sciences [96]. For the qualitative analysis, the case analyses are time-consuming, and the results can only be generalized in a limited way to the larger population. As a result, the smaller sample size raises concerns about the qualitative research's ability to generalize to the entire population [96]. In terms of generalizability, qualitative studies will never attain the same levels of sophistication as quantitative studies). The Tobit regression model is used to examine the key factors influencing the sustainability (intensity) of e-commerce. The Wilcoxon signed-rank test on a single sample is used to identify statistically significant obstacles hindering e-commerce sustainability (intensity).

3.2.1. Tobit Regression Model

A significant criticism of previous studies of e-commerce is that most of them used a binary outcome for a firm's e-commerce adoption, that is, either adopt or not adopt e-commerce. The drawback of such a binary outcome is that it does not adequately inform the technology issue [25–27]. The binary outcome cannot determine the levels of e-commerce sustainability (intensity), resulting in imprecise findings in studies on e-commerce adoption. In addition, improving the economic aspect of e-commerce sustainability results from more online shopping customers [24]. This will lead to more e-commerce utilization/intensity of Thai MSMEs. Therefore, this study uses a continuous dependent variable, which is defined as the percentage of e-commerce sales relative to total sales, as a continuous dependent variable that can capture the full range of e-commerce adoption. This variable is defined as the e-commerce sustainability in this study.

The Tobit model can capture relatively large numbers of observations at zero percent of the possible range of e-commerce adoption values in the survey of this study. Excess zeros and values bound within a given range are common data characteristics of strategic decisions, complicating the use of the linear econometric technique [97]. For example, OLS will produce biased estimators for such data with a limited dependent variable with excess zeros and values between 0 and 100. However, using SEM, a continuous dependent variable whose values are limited to 0 to 100 cannot be purged of extra zeros. Hence, a maximum likelihood estimation for a left-censored Tobit model with continuous endogenous dependent variables is well-suited for this study's data since the dependent variable's values can be zero. The ordinary least squares (OLS) method would result in biased and inconsistent estimators. This is because it would treat zero as an actual value rather than the lower limit of e-commerce adoption [98].

When the dependent variable indicating either "adoption in e-commerce" or "non-adoption in e-commerce" is censored, the OLS regression provides inconsistent estimates of the parameters, indicating that the OLS analysis coefficients will not necessarily approach the actual population parameters as the sample size increases [98,99]. To solve the issue, the left-censored Tobit model is applied, which can be expressed as follows in empirical models:

$$
\begin{aligned}
E-commerce\ &sustainability (intensity)_i^* \\
&= \beta_0 + \beta_1\ \text{computer}_i + \beta_2\ \text{smartphone}_i + \beta_3 \text{website}_i + \beta_4\ \text{owner IT skills}_i + \beta_5\ \text{owner education}_i \\
&+ \beta_6\ \text{owner gender}_i + \beta_7\ \text{owner age}_i \\
&+ \beta_8 export_i + \beta_9\ \text{B2B e} - \text{commerce}_i + \beta_{10}\ \text{firm size}_i + \beta_{11}\text{firm age}_i + \beta_{12}\text{e} - \text{commerce year}_i \\
&+ \beta_{13}\text{e} - \text{marketplaces}_i + \beta_{14}\text{social media}_i + \beta_{15}\ \text{food delivery platform}_i + \beta_{16}\ \text{retail}_i + u_i
\end{aligned}
\tag{1}
$$

where:

$$\text{E} - \text{commerce sustainability(intensity)}_i$$
$$= \begin{cases} \text{e} - \text{commerce sustainability(intensity)}_i^* \ \ if \ \text{e} - \text{commerce sustainability(intensity)}_i^* > 0 \\ \qquad\qquad 0 \qquad\qquad\quad if \ \text{e} - \text{commerce sustainability(intensity)}_i^* \leq 0 \end{cases} \quad (2)$$

where $\text{E} - \text{commerce sustainability (intensity)}_i^*$ is an unobserved (latent) variable, $\text{E} - \text{commerce sustainability(intensity)}$ is an observed variable, and $\text{E} - \text{commerce sustainability(intensity)}_L$ is the lower limit of the censored distribution. In this study, $\text{E} - \text{commerce sustainability(intensity)}_L$ is censored at 0 ($\text{E} - \text{commerce sustainability(intensity)}_L = 0$). From Equation (1), B2B e-commerce is assumed to be correlated with exports, due to the literature review in Section 2.2.2. However, B2B e-commerce directly affects e-commerce sustainability. Therefore, adopting B2B e-commerce as the instrumental variable for exports might not satisfy the exclusion restriction criteria. Hence, B2B e-commerce is included in Equation (1) to solve the omitted variable bias, leading to an unbiased estimation in this study. Descriptive statistics of the variables used in Equation (1) are given in Tables 2 and 3.

**Table 2.** Variables, definition, and summary statistics (continuous variables).

| Variable | Definition | N | Mean | SD | Min | Max |
|---|---|---|---|---|---|---|
| firm size$_i$ | The size of the company, as measured by the number of employees. | 305 | 15.227 | 24.182 | 0 | 100 |
| firm age$_i$ | A firm's age is determined by the number of operational years. | 307 | 7.557 | 8.104 | 1 | 49 |
| owner age$_i$ | Age of firm's CEO/owner/senior manager. | 300 | 40.403 | 11.332 | 17 | 67 |
| export$_i$ | A firm's e-commerce exports as measured by the proportion of e-commerce exports to total e-commerce sales. | 307 | 0.827 | 6.812 | 0 | 90 |
| e-commerce year$_i$ | The number of years a company has engaged in e-commerce. | 306 | 1.775 | 2.943 | 0 | 27 |
| B2B e-commerce$_i$ | A firm's business-to-business (B2B) e-commerce, as measured by the proportion of B2B e-commerce sales to total e-commerce sales. | 307 | 8.014 | 16.971 | 0 | 95 |
| E-commerce sustainability$_i$ | E-commerce utilization (intensity) of a company, as determined by the proportion of e-commerce sales to total sales. | 305 | 15.227 | 24.182 | 0 | 100 |

Source: Authors' calculation.

**Table 3.** Variables, definition, and summary statistics (categorical variables).

| Variable | Definition | N | No. of "1" | No. of "0" | Min | Max |
|---|---|---|---|---|---|---|
| owner gender$_i$ | A dummy variable contains the value of one if a company's CEO/owner/senior manager is male and a value of zero otherwise. | 306 | 122 | 184 | 0 | 1 |
| owner IT skills$_i$ | A dummy variable contains the value of one if a company's CEO/owner/senior manager has a degree in science, information technology (IT), or engineering, and a value of 0 otherwise (business administration, other social science and humanities, and related fields). | 307 | 28 | 279 | 0 | 1 |
| owner education$_i$ | A dummy variable is assigned a value of one if a company's CEO/owner/senior manager holds a bachelor's degree or higher and a value of 0 otherwise. | 307 | 221 | 86 | 0 | 1 |
| computer$_i$ | A dummy variable has the value of one if the business has computers (PC/laptops/tablets) and 0 otherwise. | 307 | 135 | 172 | 0 | 1 |

**Table 3.** *Cont.*

| Variable | Definition | N | No. of "1" | No. of "0" | Min | Max |
|---|---|---|---|---|---|---|
| smartphone$_i$ | A dummy variable is assigned the value one if a company has smartphones and 0 otherwise. | 307 | 181 | 126 | 0 | 1 |
| website$_i$ | A dummy variable is assigned the value of one if a business has websites or company applications and the value of 0 otherwise. | 307 | 51 | 256 | 0 | 1 |
| e-marketplaces$_i$ | A dummy variable is assigned the value of one if a company utilizes e-marketplaces (such as Lazada and Shopee) as one of its e-commerce platforms and the value of zero otherwise. | 307 | 50 | 257 | 0 | 1 |
| social media$_i$ | A dummy variable is assigned the value of one if a company uses social media (e.g., Facebook, Instagram, Line, and WhatsApp) as one of its e-commerce platforms and the value of zero otherwise. | 307 | 252 | 55 | 0 | 1 |
| food delivery platforms$_i$ | A dummy variable is one if a company utilizes social media (such as Glabfood, Foodpanda, Lineman, and Lalamove) as one of its e-commerce platforms, and the value of zero otherwise | 307 | 76 | 231 | 0 | 1 |
| retail$_i$ | A dummy variable provides a value of 1 if a company operates in the retail industry and the value of zero otherwise. | 307 | 177 | 130 | 0 | 1 |

Source: Authors' calculation.

### 3.2.2. Identifying the Barriers That Hinder E-Commerce Utilization of Thai MSMEs

According to the TOE framework, as indicated in Figure 3, the three main perceived barriers, such as (i) technological barriers, (ii) organizational barriers, and (iii) environmental barriers, are examined in this study. This study identifies eighteen barrier items based on the TOE framework (see Figure 3 and Appendix A). The One-Sample Wilcoxon Signed Rank Test is employed to examine which of the perceived barrier items significantly differs from the median scores of all perceived barriers. Unlike the One-Sample t-test, the One-Sample Wilcoxon signed-rank test is a non-parametric test, which can be used when the variable is not normally distributed. It can be used to determine whether or not the sample median equals a known expected value.

The test variable (each perceived barrier item) is compared against a hypothesized value of the sample median (the median scores of all perceived barrier items). The null hypothesis ($H_0$) and the two-tailed alternative hypothesis ($H_1$) of this test can be given as:

$H_0$: $M = M_0$; the sample's median equals the hypothesized value (the median score of all perceived barrier items).

$H_1$: $M = M_0$; the sample's median equals the hypothesized value (the median score of all perceived barrier items).

The study also evaluates the eighteen barrier items within the three main barriers. A Likert scale ranging from 0 to 5 was used to evaluate their influence on the e-commerce sustainability of Thai MSMEs, as measured by the levels of e-commerce utilization. The median score of all barrier items is 3 out of 5, indicating a moderate barrier to e-commerce utilization among Thai MSMEs.

### 4. Results

From Table 4, the F-tests of the overall significance in this study revealed a better fit than a model with only an intercept term due to the rejection of the null hypothesis ($p$-value < 0.05). In addition, the heteroskedasticity-robust standard error was used in this study, as shown in Table 4.

**Table 4.** Factors influencing the e-commerce sustainability (intensity) of Thai MSMEs.

| E-Commerce Intensity | Total Industries | | Food and Beverage | | Retail | |
|---|---|---|---|---|---|---|
| | Coef. | Robust Std. Err. | Coef. | Robust Std. Err. | Coef. | Robust Std. Err. |
| Internal e-commerce tools: | | | | | | |
| Computer | 3.2822 | (4.6345) | 4.7540 | (5.9359) | 1.7849 | (6.0207) |
| Smartphone | 16.0294 *** | (4.8592) | 7.4709 | (5.4626) | 24.1545 *** | (7.3618) |
| Website | 13.5394 ** | (5.8752) | 14.2568 * | (7.7404) | 14.6832 * | (8.0998) |
| Owner characteristics: | | | | | | |
| Owner IT skills | 2.6727 | (5.8112) | 10.3450 | (11.4364) | −1.7519 | (7.3412) |
| Owner education | −5.7135 | (5.5357) | −10.8935 | (7.4232) | −3.2253 | (8.0216) |
| Owner gender | −0.6008 | (4.0804) | 2.4625 | (5.2153) | 0.3666 | (5.5865) |
| Owner's age | −0.4416 ** | (0.2149) | −0.1669 | (0.2406) | −0.6313 ** | (0.3075) |
| Firm characteristics: | | | | | | |
| Export | 0.5434 *** | (0.0970) | 3.0954 | (2.6261) | 0.5613 *** | (0.1236) |
| B2B e-commerce | 0.4293 *** | (0.1224) | 0.5966 * | (0.3104) | 0.3659 ** | (0.1469) |
| Firm size | 0.0197 | (0.1033) | 0.1604 ** | (0.0764) | −0.1176 | (0.0744) |
| Firm age | −0.8316 *** | (0.2712) | −0.3909 | (0.4372) | −0.9625 *** | (0.2796) |
| E-commerce year | 4.0331 *** | (0.8653) | 2.5730 ** | (0.9919) | 4.5397 *** | (1.1604) |
| External e-commerce platforms: | | | | | | |
| E-marketplace | 10.1558 * | (5.4234) | 7.7694 | (5.2476) | 10.3239 | (6.8411) |
| Social media | 26.5062 *** | (6.7766) | 20.7845 *** | (7.5688) | 24.4240 ** | (10.2347) |
| Food delivery platforms | 15.5273 ** | (6.3760) | 9.7457 * | (5.8952) | | |
| Retail | 16.7139 *** | (6.2251) | | | | |
| Constant | −36.2789 *** | (13.2509) | −38.3800 ** | (16.5093) | −13.6387 | (19.8490) |
| /sigma | 26.9972 *** | (1.8085) | 20.8014 *** | (3.1978) | 28.2277 *** | (2.1693) |
| Left-censored obs. | 139 | | 75 | | 64 | |
| Uncensored obs. | 157 | | 48 | | 109 | |
| Right-censored obs. | 0 | | 0 | | 0 | |
| F statistics | 16.41 | | 3.53 | | 12.75 | |
| Prob > 0 | 0.0000 | | 0.0001 | | 0.0000 | |
| Log pseudolikelihood | −791.1206 | | −241.4363 | | −540.642 | |
| Number of obs. | 296 | | 123 | | 173 | |
| Pseudo R2 | 0.1174 | | 0.1167 | | 0.1101 | |

Note: *** indicates a 1% level of significance; ** indicates a 5% level of significance, * indicates a 10% level of significance.

*4.1. Technological Factors: Firms' Internal E-Commerce Tools*

A firm's existing and new technologies can affect the e-commerce intensity of MSMEs. In this study, technological factors include e-commerce tools such as computers, smartphones, and websites. For internal e-commerce tools, as indicated in Table 4, smartphones are significantly and positively associated with the e-commerce intensity of Thai MSMEs. The same result is obtained when the sample is disaggregated for retail services. However, computers are not statistically significant for increasing e-commerce intensity. For the firms' internal e-commerce tools, websites are significantly and positively related to the e-commerce intensity of Thai MSMEs. The same result is obtained when the sample is disaggregated for retail and food and beverage service.

*4.2. Organizational Factors*

4.2.1. Owner's (CEO) Characteristics

This study found that the owner's age has a significant and negative impact on the e-commerce intensity of Thai MSMEs. Nevertheless, the owner's (CEO) IT skills, education, and gender do not significantly affect the e-commerce intensity of Thai MSMEs. These findings imply no statistical difference in the e-commerce intensity among owners (CEOs) with at least a bachelor's degree compared with those who did not have such a degree. In addition, there is no statistical difference in the e-commerce intensity between owners (CEOs) with education degrees in science, IT, and engineering and those owners (CEOs)

with education in accounting, business administration, economics, social science, and humanities and related fields. There is no statistical difference in the e-commerce intensity between female owners (CEOs) and male owners (CEOs). Finally, there is statistical insignificance in e-commerce intensity between owners (CEOs) with at least a bachelor's degree and those without a bachelor's degree.

4.2.2. Exports and Business to Business (B2B)

Focusing on all Thai MSMEs, the findings suggest that exports tend to increase the e-commerce sustainability of the firms in the sample. The same result is obtained when the sample is disaggregated for retail services. This result implies that firms with exporting experience are likely to adopt resources and capabilities to improve their business operations and technology for e-commerce activities. A significant and positive result is also obtained when the sample is disaggregated for retail services. This result implies that exporting firms can gain from learning by exporting, due to knowledge transfers from foreign counterparts. Thus, this study confirms H3, in that e-commerce exports positively affect the e-commerce sustainability of Thai MSMEs.

From Table 4, this study further indicates a significant and positive association between B2B e-commerce and e-commerce sustainability. The significant and positive finding is also obtained when the sample is disaggregated for retail and food and beverage services. This finding is consistent with Amornkitvikai and Tangpoolcharoen [73], Wicaksono [72], and ETDA [16], where the statistics showed that B2B e-commerce plays a significant role in supply chains and dominates e-commerce transactions in Thailand. It also confirms H2, in that B2B e-commerce positively affects the e-commerce sustainability of Thai MSMEs.

4.2.3. Firm Size, Firm Age, and Firm E-Commerce Experience

Firm size is not found to influence the e-commerce intensity significantly for the sampled respondents. The same result is also obtained when the sample is disaggregated into F&B and retail services. Nevertheless, this study confirms that firm age significantly and negatively affects the e-commerce intensity of Thai MSMEs, as shown in Table 4. Furthermore, firms with more e-commerce experience positively and significantly affect the e-commerce intensity of Thai MSMEs since learning by doing experience in e-commerce can enhance their e-commerce intensity. When disaggregated by sub-sectors, it is found that it can significantly increase e-commerce intensity for retail and F&B services.

*4.3. Environmental Factors (External E-Commerce Platforms)*

Environmental factors are related to external factors that can significantly influence the e-commerce intensity of Thai MSMEs. External e-commerce platforms such as social media (i.e., Facebook, Instagram, Twitter, Line, and WhatsApp), e-marketplaces (i.e., Lazada and Shoppee), and food delivery platforms (i.e., Lineman, Glabfood, Lalamove, and Foodpanda) are significantly and positively related to the e-commerce intensity of Thai MSMEs. Specifically, social media plays the most significant role in promoting e-commerce intensity due to the magnitude of the estimated coefficient of social media, as it is the largest coefficient.

*4.4. Identifying the Barriers That Hinder the E-Commerce Sustainability of Thai MSMEs*

According to the TOE framework, the study also evaluates the eighteen barrier items within each of the four main barriers. A Likert scale ranging from 0 to 5 was used to evaluate their influence on the e-commerce utilization of Thai MSMEs. The median score of all barrier items is 3 out of 5, indicating a moderate barrier to e-commerce utilization among Thai MSMEs, as shown in Table 5. Unlike the One-Sample *t*-test, the One-Sample Wilcoxon signed-rank test, a non-parametric test, is suitable in this study since all 18 barrier items are not normally distributed due to the statistical significance of the Kolmogorov–Smirnov Normal Test, as revealed in Table 5.

**Table 5.** Barriers that hinder the levels of e-commerce utilization of Thai MSMEs: Reliability Test, One-Sample Wilcoxon Signed Rank Test, and One-Sample Kolmogorov–Smirnov Normal Test.

| Barrier Factors | Reliability Test | One-Sample Wilcoxon Signed Rank Test (Test Value = 3) | | | | | | One-Sample Kolmogorov–Smirnov Normal Test |
|---|---|---|---|---|---|---|---|---|
| | Cronbach's Alpha if Item Deleted | Mean | Std. Dev. | Total N | Test Statistics | Std. Error. | Standardized Test Statistic | Asymptotic Sig (2-Sided Test) | Test Statistic (Asymptotic Sig) |
| Technological Barriers | | | | | | | | |
| TechI: The company's technological infrastructure (including its website) does not support e-commerce. | 0.808 | 2.87 | 0.935 | 307 | 4265.000 | 505.625 | −2.764 | 0.006 *** | 0.290 (0.000) *** |
| TechII: E-commerce activities (e.g., marketing, payment, logistics) remain segmented. | 0.806 | 3.22 | 0.839 | 307 | 23,345.000 | 1529.966 | −0.192 | 0.848 | 0.234 (0.000) *** |
| TechIII: Inadequate security for online transactions and payments. | 0.804 | 3.34 | 1.027 | 307 | 26,545.000 | 1540.248 | 1.887 | 0.059 * | 0.225 (0.000) *** |
| TechIV: Inadequate security measures to prevent hacking and malware. | 0.801 | 3.59 | 0.940 | 307 | 33,397.000 | 1537.144 | 6.348 | 0.000 ** | 0.219 (0.000) *** |
| Organizational Barriers | | | | | | | | |
| OrgI: The company's product or service is not suitable for online transactions. | 0.807 | 2.81 | 1.044 | 307 | 12,842.000 | 1538.297 | −7.019 | 0.000 ** | 0.228 (0.000) *** |
| OrgII: Lack of technical understanding or awareness of available e-commerce training. | 0.801 | 3.29 | 0.927 | 307 | 25,280.000 | 1536.391 | 1.068 | 0.285 | 0.206 (0.000) *** |
| OrgIII: The company's size is insufficient to support e-commerce activities. | 0.798 | 3.17 | 1.059 | 307 | 22,355.000 | 1541.596 | −0.833 | 0.405 | 0.204 (0.000) *** |
| OrgIV: Financial/human resource constraints prevent investment in e-commerce activities. | 0.801 | 3.09 | 1.013 | 307 | 20,703.000 | 1536.715 | −1.911 | 0.056 * | 0.235 (0.000) *** |
| CEOI: Uncertainty over the proportion of e-commerce benefits to expenses. | 0.800 | 3.31 | 0.895 | 307 | 25,776.000 | 1533.951 | 1.393 | 0.164 | 0.217 (0.000) *** |
| CEOII: Still acquiring knowledge of e-commerce transactions and markets. | 0.811 | 3.45 | 0.852 | 307 | 30,742.000 | 1531.125 | 4.639 | 0.000 *** | 0.239 (0.000) *** |
| CEOIII: Concerning the management of disruptive e-commerce technologies. | 0.811 | 3.36 | 0.890 | 307 | 27,701.000 | 1531.102 | 2.653 | 0.008 *** | 0.223 (0.000) *** |
| Environmental Barriers | | | | | | | | |
| EnvironI: My country's telecommunications and other logistics infrastructure are inadequate. | 0.798 | 2.94 | 0.941 | 307 | 15,310.000 | 1532.176 | −5.436 | 0.000 *** | 0.258 (0.000) *** |
| EnvironII: Governmental incentives are inadequate. | 0.801 | 3.47 | 0.860 | 307 | 31,130.000 | 1531.776 | 4.890 | 0.000 *** | 0.225 (0.000) *** |
| EnvironIII: Insufficiently qualified vendors for website development and maintenance. | 0.797 | 3.27 | 0.891 | 307 | 24,816.000 | 1532.234 | 0.768 | 0.442 | 0.226 (0.000) *** |

**Table 5.** *Cont.*

| Barrier Factors | Reliability Test | One-Sample Wilcoxon Signed Rank Test (Test Value = 3) | | | | | | | One-Sample Kolmogorov–Smirnov Normal Test |
|---|---|---|---|---|---|---|---|---|---|
| | Cronbach's Alpha if Item Deleted | Mean | Std. Dev. | Total N | Test Statistics | Std. Error. | Standardized Test Statistic | Asymptotic Sig (2-Sided Test) | Test Statistic (Asymptotic Sig) |
| EnvironIV: Absence of government standards or regulations for e-commerce activities. | 0.798 | 3.44 | 0.816 | 307 | 30,470.000 | 1528.847 | 4.468 | 0.000 *** | 0.234 (0.000) *** |
| EnvironV: A significant proportion of customers are still unfamiliar with e-commerce. | 0.800 | 3.59 | 1.000 | 307 | 33,324.000 | 1536.570 | 6.303 | 0.000 *** | 0.257 (0.000) *** |
| EnvironVI: E-payment (commissions, bank fees, etc.) and logistics incur high additional expenses. | 0.800 | 3.42 | 0.916 | 307 | 28,931.000 | 1535.755 | 3.446 | 0.001 *** | 0.212 (0.000) *** |
| EnvironVII: competitors' pressure is not enough. | 0.806 | 2.93 | 0.962 | 307 | 14,439.000 | 1533.815 | −5.998 | 0.000 *** | 0.235 (0.000) *** |

Notes: *** indicates a 1% level of significance; ** indicates a 5% level of significance, * indicates a 10% level of significance.

For technological barriers, the respondents mostly agree that the items "Inadequate security measures to prevent hacking and malware", "Inadequate security for online transactions and payments", and "the company's technological infrastructure (including its website) does not support e-commerce" are found to be statistically significant barriers, which is due to the median of the sample being statistically different from the hypothesized value of 3 (the median score of all perceived barrier items).

For organizational barriers, the barrier items "their product/service is not suitable for e-commerce transaction" and "they are constrained by financial/human resources to invest in e-commerce activities" are statistically significant in hindering the e-commerce sustainability among Thai MSMEs. As a part of organizational barriers, the perceived barriers from the CEO's beliefs also hinder their e-commerce sustainability. First, the CEOs indicated that they are still learning e-commerce transactions and e-markets, followed by their concern about managing disruptive e-commerce technologies.

Environmental barriers are likely to reduce MSMEs' e-commerce sustainability because the respondents perceived that many customers are still not literate in adopting e-commerce. It is followed by insufficient government incentives, a lack of standards and regulations from the government on e-commerce activities, insufficient government incentives, and the high cost of e-payment and logistics. Overall, it can be summarized that environmental, technological, and CEO barriers are statistically significant barriers that deter the e-commerce sustainability of the SME respondents in the survey. As a result, this study confirms H7, in that the TOE barriers can hinder the e-commerce sustainability of Thai MSMEs.

## 5. Discussion

This study employs the Tobit regression model to examine significant factors affecting the e-commerce sustainability of Thai MSMEs. The Tobit model can handle the values of e-commerce sustainability (intensity), which consists of excess zeros and values bounded between 0 and 100. However, using SEM, the values of e-commerce intensity (a continuous dependent variable whose values are limited to 0 to 100) cannot be purged of extra zeros. Due to the non-normality of all 18 barrier items, the One-Sample Wilcoxon Signed Rank Test, a non-parametric test, is employed to examine which of the perceived barrier items significantly differs from the median scores of all perceived barriers.

The empirical results in Tables 4 and 5 theoretically and practically confirm the validation of the TOE framework, which is more oriented to firm-level analysis. Based on technological factors, the positive evidence between smartphones and e-commerce intensity implies that Thai MSMEs can adopt smartphones as one of the technological tools to promote their e-commerce intensity. This result is similar to Apergis [46], suggesting that younger entrepreneurs prefer to use smartphones to run their businesses. Hence, smartphones are essential for conducting today's online businesses. However, computers are not statistically significant for increasing e-commerce intensity, implying that smartphones can replace personal computers in e-commerce. This evidence is inconsistent with Walker, Saffu, and Mazurek [41], who showed that PC networks are statistically different for e-commerce adopters and non-adopters. Websites are found to increase e-commerce intensity for Thai MSMEs. This positive finding is similar to that of Salehi, Abdollahbeigi, Langroudi, and Salehi [48] and Walker, Saffu and Mazurek [41], and Rahayu and Day [40], suggesting that websites with clear information about firms' products and services can increase their e-commerce transactions. The result also confirms the Rotchanakitumnuai and Speece [52] findings stating that Thai firms have adopted websites to sell online. Hence, these results confirm H1, in that a firm's e-commerce tools positively affect the e-commerce intensity of Thai MSMEs.

Focusing on organizational factors, the significant and negative effect of the owner's age on the e-commerce intensity of Thai MSMEs implies that older owners (CEOs) are likely to be less intense in their e-commerce usage than their younger counterparts. Hence, the evidence confirms H2.4, in that the owner's age negatively affects the e-commerce sustainability of Thai MSMEs. It also confirms the findings of Nair, Chellasamy, and Singh [57], Muathe and Muraguri-Makau [54], Chuang, Dwivedi, Nakatani, and Zhou [58]. Nevertheless, the owner's (CEO) IT skills, education, and gender do not significantly affect the e-commerce intensity of Thai MSMEs. As a result, these findings confirm H2.4 but fail to confirm H2.1, H2.2, and H2.3.

Exports can help increase their e-commerce intensity, focusing on all Thai MSMEs, including those in retail services. This result is similar to Kraemer, Gibbs, and Dedrick [64] and Terzi [63], implying that firms with exporting experience will likely adopt resources and capabilities to improve their business operations and technology for e-commerce activities. In addition, exporting firms can gain from learning by exporting due to knowledge transfers from foreign counterparts. Thus, this study confirms H3, in that e-commerce exports positively affect the e-commerce intensity of Thai MSMEs. In addition, B2B e-commerce positively influences Thai MSMEs' e-commerce intensity, including those in retail and food and beverage services. This finding is consistent with Amornkitvikai and Tangpoolcharoen [73], Wicaksono [72], and ETDA [16], where the statistics showed that B2B e-commerce plays a significant role in supply chains and dominates e-commerce transactions in Thailand. It also confirms H2, in that B2B e-commerce positively affects the e-commerce intensity of Thai MSMEs. In addition, the e-commerce experience is also critical for enhancing their e-commerce intensity. The result of this study is different from Ramanathan, Ramanathan, and Hsiao [76], who found no significant association between e-commerce experience and e-commerce performance for Taiwanese MSMEs.

The empirical result of this study does not confirm H5.1, in that firm size positively affects the e-commerce intensity of Thai MSMEs. The empirical result of this study is also in line with the findings of Rahayu and Day [47], who found an insignificant result for MSMEs' e-commerce adoption in Indonesia. However, the result of this study is inconsistent with the findings of Lertwongsatien and Wongpinunwatana [23] and Brown and Kaewkitipong [19], which had significant and positive results.

The significant and negative association between firm age and e-commerce intensity implies that older Thai MSMEs are likely to utilize lower e-commerce intensity since they face IT implementation constraints. In other words, older MSMEs may prefer brick-and-mortar (offline) sales over online sales as they may not be familiar with newer technologies. Therefore, this result accepts H5.2, in that firm age negatively affects the e-commerce

intensity of Thai MSMEs. This evidence supports Nair, Chellasamy, and Singh [57]'s findings that firm age can inhibit organizational readiness.

For environmental factors, external e-commerce platforms such as social media, e-marketplaces, and food delivery platforms can be e-commerce platforms that improve the e-commerce intensity of Thai MSMEs. More importantly, social media are the essential e-commerce platform in promoting e-commerce intensity. The top three social commerce sites in Thailand are Facebook, Instagram, and Line, while the most attractive channel is Facebook [100]. A significant and positive result for social media is also obtained when the sample is disaggregated into F&B and retail services. Thus, e-commerce platforms can help Thai MSMEs reach greater e-commerce intensity levels. This result confirms H6, in that external e-commerce platforms positively affect the e-commerce intensity of Thai MSMEs. Finally, Thai MSMEs in the retail industry are likely to have greater e-commerce intensity than those in the F&B services industry due to a significant and positive estimated coefficient in the industry (retail) variable. The econometric findings confirm H1, H2.4, H3, H4, H5.2, H5.3, and H6. However, they do not confirm H2.1, H.2.2, H2.3, and H5.1.

Lastly, focusing on the TOE framework, the non-parametric results of the One-Sample Wilcoxon Signed Rank Test indicate that thirteen out of eighteen barrier items can significantly impede the e-commerce sustainability of Thai MSMEs, confirming H7. Thai MSMEs perceive that environmental barriers mainly hinder their e-commerce sustainability, since six out of seven environmental barriers are statistically different from their median barrier scores. For example, many customers are still not literate in adopting e-commerce. They also perceive that Thailand still lacks security to prevent hacking and malware. In addition, technical barriers also hinder e-commerce sustainability, since three out of four technological barriers are statistically different from their median barrier scores. For example, the respondents mostly agree that e-commerce security is insufficient to prevent hacking and malware. In addition, they perceive insufficient security for online payments and transactions, and their technological infrastructures do not support their e-commerce business.

Lastly, organizational barriers impede e-commerce sustainability, since four out of seven organizational barriers are statistically different from their median barrier scores. For instance, Thai MSEMEs perceive their products or services as unsuitable for e-commerce transactions. Thai MSMEs perceive that they are constrained by financial and human resources to invest in e-commerce activities. In addition, Thai CEOs perceive that their knowledge of adopting e-commerce is still limited, and they are still learning how to cope with e-commerce transactions and markets. They are concerned about managing disruptive e-commerce technologies.

## 6. Theoretical and Practical Implications

### 6.1. Theoretical Implication

The findings of this study validate the technology–organization–environment (TOE) framework for Thai SMEs in the retail and food and beverage services sectors. This study confirms the importance of technological factors, as firm websites and smartphones are essential e-commerce tools for increasing the e-commerce intensity of Thai MSMEs. Furthermore, this study confirms the significance of organizational factors, as particular firm and owner characteristics substantially impact the sustainability of e-commerce. The results also demonstrate the significance of the environmental readiness of Thai SMEs. External e-commerce platforms such as social media, e-marketplaces, and food delivery platforms offered by social media networking and online platform providers can be viewed as the intermediaries between Thai MSMEs and customers. Table 5 reveals that the TOE framework can be used to identify significant barriers impeding the e-commerce sustainability (intensity) of Thai MSMEs, covering 13 out of 18 barrier items.

The TOE framework, which is more oriented toward firm-level analysis than other theories such as the technology adoption model (TAM) and the diffusion of innovation theory (DOI), is applied to this study. In addition, the TOE framework, similar to the DOI

theory, can encompass some contextual factors, such as the impact of mass media communication channels and the innovative decisions resulting from the personal characteristics of business owners (CEOs). Consequently, this can boost innovation diffusion (or e-commerce intensity) among Thai MSMEs. Significantly, the inclinations of Thai CEOs or business owners can increase the e-commerce intensity of Thai MSMEs, as highlighted by the DOI theory of technology diffusion.

*6.2. Practical Implication*

The results of this study have several practical implications for promoting the e-commerce sustainability of Thai MSMEs. Due to the importance of external e-commerce platforms for e-commerce sustainability, the government should promote more accessible, affordable, and reliable e-commerce platforms for Thai MSMEs. In addition, local e-commerce platforms should be promoted to prevent the domination of foreign e-commerce platforms. The government should also promote Thailand's sustainable e-commerce ecosystems and significantly push for a more digital-friendly ecosystem due to this study's significant findings on internal e-commerce tools and external e-commerce platforms. Promoting the effective use of technological tools and reliable e-commerce platforms needs strong upstream-downstream linkages of e-commerce transactions in the country. For instance, good infrastructure and logistics should be promoted to support increasing e-commerce transactions for Thai MSMEs. Furthermore, high internet speeds and more advanced information communication technology (ICT) tools should be promoted to increase e-commerce transactions.

This study pointed out that older owners (CEOs) and firms are likely to adopt lower e-commerce sustainability since they may lack the IT knowledge and e-commerce literacy necessary for adopting e-commerce. Hence, IT and e-marketing knowledge should be targeted to older owners to adopt and increase their e-commerce. In addition, more e-commerce experience is found to increase the e-commerce sustainability of Thai MSMEs. Therefore, public e-commerce knowledge and one-stop service counseling centers should be established to promote comprehensive knowledge of e-commerce for Thai MSMEs, especially for the less experienced and new e-commerce users who are MSMEs.

For F&B services, online food delivery platforms have become popular among Thai customers in the Bangkok Metropolitan Region. Therefore, the government should encourage local food delivery platforms to avoid dominating international food delivery platforms (i.e., Lineman, Glabfood, Lalamove, and Foodpanda). The domination of international food delivery platforms can result in a very high gross profit (GP) charge. As a result, promoting more competition of food delivery platforms can reduce Thai MSMEs' operational costs by paying lower GP charges, leading to higher levels of e-commerce sustainability. These can help promote Thai MSMEs' e-commerce sustainability, thereby creating more jobs and stimulating local economic development and sustainability. Finally, exports and B2B e-commerce significantly increase e-commerce sustainability. Therefore, promoting cross-border e-commerce should be prioritized for Thai MSMEs keen to internationalize their operations.

Based on obstacles to e-commerce sustainability, Thai MSMEs should be provided with more accessible, affordable, and secure e-commerce platforms. In particular, the government should vigorously enforce all laws pertaining to illegal e-commerce transactions. In the Bangkok Metropolitan Region, food delivery platforms have gained popularity among Thai customers. However, food delivery services are restricted to the Bangkok Metropolitan area and a handful of large provinces (or tourist cities), including Chiang Mai, Kon Kaen, Pattaya, and Hat Yai [101]. Therefore, the government of Thailand should promote online food delivery services throughout all regions. This can facilitate the expansion of e-commerce transactions for Thai MSMEs, resulting in the creation of additional jobs and the growth of the local economy. In addition, technological infrastructure must be bolstered to accommodate the growth of e-commerce transactions.

Thai entrepreneurs, especially the older ones, should be provided with more e-commerce literacy programs. The government can offer regular online and offline e-commerce training programs for Thai entrepreneurs in all regions of Thailand. In addition, customers in Thailand should be educated about e-commerce via online and offline training programs emphasizing online purchases' security. Thai entrepreneurs believe that Thai consumers are not yet sufficiently educated to use e-commerce.

Lastly, Thai e-commerce MSMEs, particularly those in the retail sector, should prioritize improved infrastructure, improved logistics, sufficient financial resources, high internet speed, more advanced ICTs, sufficient government support, and secure payment systems.

## 7. Conclusions

This study uses the left-censored Tobit regression model to investigate critical factors influencing the e-commerce sustainability of Thai MSMEs. Unlike SEM, the Tobit model can handle excess zeros and values bounded between 0 and 100. In addition, the One-Sample Wilcoxon Signed Rank Test, a non-parametric test, is used to examine significant barriers hindering their e-commerce sustainability. This study's survey is based on the TOE framework, which includes technological, organizational, and environmental variables with CEO characteristics included in the organizational factors to take into account the decision-making structures of Thai MSMEs, where the CEO/owner are frequently the sole decision makers.

Based on technological factors, unlike personal computers (PCs), smartphones can be an effective e-commerce tool to help Thai MSMEs increase their e-commerce sustainability. From the interviews, Thai entrepreneurs in the retail industry mostly use their own Facebook pages or Instagram via their smartphones to conduct online sales. These results imply that computers are no longer a powerful e-commerce tool in conducting e-commerce business and will likely be replaced by smartphones. Websites can significantly increase the e-commerce sustainability of Thai MSMEs.

The crucial finding for organizational factors highlights that exports significantly and positively influence Thai MSMEs' e-commerce sustainability due to the learning-by-exporting experience. In addition, B2B e-commerce can be crucial in significantly driving e-commerce sustainability since B2B e-commerce dominates Thailand's e-commerce and plays a significant role in supply chains, eventually increasing e-commerce sustainability. Moreover, older firms are less likely to adopt e-commerce sustainability than younger counterparts due to the incapability of adopting new technologies. However, firms with more e-commerce experience have higher e-commerce sustainability due to the cumulative learning-by-doing knowledge.

Based on CEO owner characteristics, this study confirms that older CEO owners tend to use e-commerce less intensively than younger CEO owners, possibly because they are unfamiliar with or less capable of using newer technologies. Therefore, they are more likely to open physical stores than click-and-mortar stores. In addition, this study found that the owner's (CEO's) educational attainment, education degrees in science, information technology (IT), and engineering, as well as the owner's (CEO's) gender, do not significantly impact the e-commerce sustainability of Thai MSMEs.

Regarding environmental factors, this study revealed that external e-commerce platforms such as social media (i.e., Facebook, Instagram, Line, and WhatsApp), e-marketplaces (i.e., Lazada and Shoppee), and food delivery platforms (i.e., GrabFood, Line Man, Lalamove, and Foodpanda) could significantly increase e-commerce sustainability. Therefore, social media plays the most significant role in promoting e-commerce sustainability since it is easily accessible with low barriers to entry and minimizes costs. Moreover, Thai MSMEs do not need high IT skills to access it. As a result, most Thai MSMEs are likely to adopt social media to sell online products in the retail and food and beverage industries. Furthermore, according to ETDA [16], Thai MSMEs will likely adopt Line, Facebook, and Instagram as three main online marketing channels. Finally, the significant and positive

result of food delivery platforms implies that Thai customers favor purchasing food and beverages online due to the convenience accorded by these platforms.

This study also confirms that TOE barriers significantly hinder the e-commerce viability of Thai MSMEs. Thirteen of the eighteen obstacles reported by Thai SMEs are statistically different from the median score of all eighteen obstacles, indicating that thirteen obstacles pose a significant hindrance to their e-commerce sustainability. Furthermore, six out of seven environmental obstacles have scores statistically different from their corresponding medians. For example, many customers are still illiterate when it comes to e-commerce. In addition, they believe Thailand has insufficient security measures to prevent hacking and malware. In addition, technical barriers hinder the sustainability of e-commerce because three out of four technological barriers statistically differ from their median barrier scores. For instance, most respondents concur that e-commerce security is inadequate to prevent hacking and malware. In addition, they believe there is insufficient security for online transactions and payments, and their technological infrastructure is inadequate to support their e-commerce business. Finally, four of the seven organizational barriers are statistically and noticeably different from their respective median barrier scores. This makes it hard for them to keep up with e-commerce.

## 8. Limitation and Future Studies

The TOE framework is more oriented toward firm-level analysis. However, due to the limited number of questions available in the questionnaire survey, other essential variables for the TOE framework, such as other technologies available to a firm, industry characteristics and market structures, government regulations, internal factors of a firm (i.e., financial position, top management support), and other organizational factors (i.e., pressures arising from competitors, consumers, and suppliers; reliability), can be included in future studies. Future research, which focuses on both the supply and demand sides of e-commerce adoption and intensity, could be examined. In addition, other theoretical frameworks can be applied for future studies, such as the diffusion of innovation theory (DOI), the technology adoption model (TAM), and the planned behaviors (TPB) theory. Extending new research in these directions can further enhance the understanding of e-commerce adoption for Thai MSMEs and online buyers, leading to better policy formulation for the future. In addition, a significant increase in e-commerce has been observed since the COVID-19 pandemic, particularly in the F&B and retail sectors. However, it is outside the scope of this study. Furthermore, Thai MSMEs are likely to adopt e-commerce more frequently to avoid bankruptcy during lockdown periods. The period since the COVID-19 pandemic should therefore be considered in future research. Lastly, future research may employ a mixed-method approach (quantitative and qualitative analyses). As a result, the quantitative analysis has not been able to obtain in-depth information; instead, it has provided an overview of the study.

**Author Contributions:** Conceptualization, S.Y.T. and Y.A.; methodology, Y.A.; software, Y.A.; validation, C.H. and W.W.B.; formal analysis, Y.A.; investigation, Y.A.; data curation, Y.A.; writing—original draft preparation, Y.A.; writing—review and editing, S.Y.T., C.H. and W.W.B.; visualization, W.W.B.; supervision, S.Y.T. and C.H.; project administration, Y.A.; funding acquisition, S.Y.T. All authors have read and agreed to the published version of the manuscript.

**Funding:** The survey questionnaire of this research was funded by ISEAS-Yusof Ishak Institute, Singapore; grant number [220618].

**Acknowledgments:** The authors are grateful to the four anonymous referees, the academic editor, and the journal's assistant editor for their insightful comments and suggestions to improve the quality of this article. We would also like to thank Cassey Lee and Jiraporn Tangpoolcharoen for their insightful comments and suggestions on this research project.

**Conflicts of Interest:** The authors declare no conflict of interest.

**Appendix A. The Structure of the Survey Questionnaire**

*Appendix A.1. Profile of the Firm*

Name of company:
Address:
Name of respondent:
Position of the respondent in the company:
Email of respondent:
Telephone number of respondent:

　　　Please tick (/) the box or fill in the blanks where appropriate

1.　In which sector/sub-sector is your firm classified: (i) Food, (ii) Beverage, (iii) Food and Beverage, and (iv) Retail.
2.　The number of full-time/permanent employees, as of the end of 2017.
3.　Year of establishment of the company.
4.　Sales revenue, as of the end of 2017.
5.　Do you use the internet in your business process?
6.　For what purpose do you use the internet:
7.　Types of technology available in the firm: Please tick (may tick more than one)
8.　What is the highest education level completed by the firm's CEO/owner/Senior Manager
9.　If the firm's CEO/owner/Senior Manager obtained at least a university degree, please tick the field:
10.　Age of CEO/owner/Senior Manager
11.　Gender of CEO/owner/Senior Manager
12.　How does your firm conduct sales/purchase: (i) only online, (ii) online and offline, and (iii) offline? Please skip question 20
13.　Have you received any incentive or grant for using e-commerce (Please circle)
14.　How long has your firm been using e-commerce?
15.　E-commerce revenue as a percentage of total sales revenue, as of the end of 2017 (%)
16.　E-commerce as a percentage of total export revenue at the end of 2017 (%) (Please put zero if there are no export activities)
17.　List export destinations by the end of 2017.
18.　E-commerce sales revenue is classified by the type of e-commerce transactions (B2B, B2C, and B2G) as of the end of 2017
19.　Please tick the following online platforms currently used in your firm to promote (for offline) and conduct e-commerce sales.
20.　Please tick the following e-commerce options:

|  | Levels of Adoption | Please Tick |
|---|---|---|
| For Ecommerce users | | |
| a. | Not interested in using any e-commerce platforms | |
| b. | I am interested in using e-commerce but do not know how to adopt it. | |
| c. | Planning to use e-commerce within the next two years | |
| For Ecommerce users | | |
| d. | Own Website, static, or there is no interaction with customers. | |
| e. | Own Website, interactive that is using it to communicate with customers | |
| f | Online payment facilities for e-commerce transactions | |
| g | Outsourced to third-party providers | |
| h | Mobile commerce (M-commerce) without an online payment system | |
| j | M-commerce with an online payment system | |

*Appendix A.2. Identifying Barriers to E-Commerce Adoption*

21.  To what extent do the following issues hinder your organization from using or using more e-commerce (Please tick only ONE appropriate score on a scale of 1 to 5)

|  |  | Strongly Disagree 1 | Somewhat Disagree 2 | Neither Disagree nor Agree 3 | Somewhat Agree 4 | Strongly Agree 5 |
|---|---|---|---|---|---|---|
| A. | Technological Factors |  |  |  |  |  |
| I | The company's technological infrastructure (including its website) does not support e-commerce. |  |  |  |  |  |
| II | E-commerce activities (e.g., marketing, payment, logistics) remain segmented. |  |  |  |  |  |
| III | Inadequate security for online transactions and payments |  |  |  |  |  |
| IV | Inadequate security measures to prevent hacking and malware |  |  |  |  |  |
| B. | Organizational Factors |  |  |  |  |  |
| I | The company's product or service is not suitable for online transactions. |  |  |  |  |  |
| II | Lack of technical understanding or awareness of available e-commerce training. |  |  |  |  |  |
| III | The company's size is insufficient to support e-commerce activities. |  |  |  |  |  |
| IV | Financial/human resource constraints prevent investment in e-commerce activities. |  |  |  |  |  |
| C. | Environmental Factors |  |  |  |  |  |
| I | My country's telecommunications and other logistics infrastructure are inadequate. |  |  |  |  |  |
| II | Governmental incentives are inadequate. |  |  |  |  |  |
| III | Insufficiently qualified vendors for website development and maintenance |  |  |  |  |  |
| IV | Absence of government standards or regulations for e-commerce activities |  |  |  |  |  |
| V | A significant proportion of customers are still unfamiliar with e-commerce. |  |  |  |  |  |
| VI | E-payment (commissions, bank fees, etc.) and logistics incur high additional expenses. |  |  |  |  |  |
| VII | Competitors' pressure is not enough. |  |  |  |  |  |
| D. | CEO/Owner/Senior Management views |  |  |  |  |  |
| I | Uncertainty over the proportion of e-commerce benefits to expenses. |  |  |  |  |  |
| II | Still acquiring knowledge of e-commerce transactions and markets. |  |  |  |  |  |
| III | Concerning the management of disruptive e-commerce technologies |  |  |  |  |  |

22.  Rank the importance of technological, organizational, environmental, and CEO/owner/Senior Manager's views in hindering eCommerce implementation, with one as the most important and four as the least important.

|  | Technological Variables | Organizational Variables | Environmental Variables | CEO/Owner/Senior Manager |
|---|---|---|---|---|
| Rank 1 = the <u>most</u> important 4 = the <u>least</u> important |  |  |  |  |

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
