# Peer review of "Barriers and Factors Affecting the E-Commerce Sustainability of Thai Micro-, Small- and Medium-Sized Enterprises (MSMEs)"

_sustainability, doi:10.3390/su14148476_

Round 1
Reviewer 1 Report
Dear authors,
I commend the authors for the extensive work put into the study, its aims and results are very timely and contribute greatly to the field of e-commerce specially with the regional perspective.
The use of TOE is well accepted research area for e-commerce sustainability but there are multiple other factors affecting the adoption of e-commerce for MSME beyond the TOE that should have been referenced.
In addition, the authors used Tobit Regression and T-Test for the analysis. I wonder why? Since this study is investigating factor interactions among the many variables, and based on that, a structural equation modeling analysis would have been the ideal?
Also the use of T-test on non-parametric data such as survey data was not advised, I didn't find the authors' justification for the use of T-test unlike other non-parametric analysis tools.
More importantly, there was not a graphic research model to ascertain the relationship categories and their directions, I’d have liked to see a clear research model depicting all the hypotheses and their relationships.
Reviewer 2 Report
Hello,
I read the study with great pleasure. Congratulations to the authors for their work. The work is clear, well organized, the hypotheses are in accordance with the existing literature. I appreciated that you allocated an important part of the paper to build the hypotheses and explain the results. The discussions and conclusions are in line with the results of previous research and studies. The title of the scientific paper agrees with the content.
Recommendation
At point 3. Data and methodology
I think it is necessary to present the questionnaire, the main categories of questions / items. A presentation of the respondents - I deduced some of the characteristics of the respondents in the "Results" section, but I think that a supplement should be made here.
At point 5/6 Results / Conclusions
Given that the data were collected in 2018-2019 and therefore does not include the Covid-19 pandemic period (2020-2021) - I think that a supplement should be made regarding this period, especially if the paper will be published in this year. At least in the second year of the pandemic, there were significant increases in e-commerce, especially in the F&B sector and not only, SMEs were the first to adapt / reorient to this sector as a "solution" to avoid bankruptcy during lockdown periods. Perhaps it should be presented as a concern for future research / limitation of the present study.
Thank you very much!
Reviewer 3 Report
Put the theoretical and managerial implication
To be honest, this manuscript goes against the norm regarding quantitative research. From what I know so far, the quantitative approach is aimed at "investigating" further the results of qualitative research. That is why quantitative methods are sometimes intended to reproduce developing or already established models.
However, the author of this manuscript tries to make something different. I think it is acceptable as long as it is supported by relevant references considering that the researcher wants to find the "barriers factor" as stated in his study.
Of course, it is undoubtedly acceptable if it is only based on a similar approach as in this manuscript. However, once again, the quantitative approach requires the support of the relevant literature to develop the model.
On the other hand, on the methodology side, I feel pretty good for now. I want to know to what extent the author can summarize the theoretical and managerial implications. After that, if the executive and theoretical implications can be understood, I will only review the methodology's deepening and discussion.
The discussion section only discusses normative things and has not been able to answer in detail the main points of the study of the problems raised in this study. However, I try to allow the author to improve the manuscript little by little, considering that the survey presented may seem quite simple in the title but complex and comprehensive if you want to study it in depth.
Reviewer 4 Report
1. Author/s should make a decision: Their study is dealing with “Micro-, Small and Medium-Sized Enterprises” (as in title) OR “Small and Medium-Sized Enterprises” (as in keywords).
2. Keywords as (“Small”, “Service”) are too general; they should be more specific. In addition, “Small” seems to be redundant (as compared to “Small and Medium-Sized Enterprises”). It is recommended to revise the Keywords list (remove “Small”, make “Services” more specific, etc.).
3. The basic prerequisite of the study seems to be “e-commerce sustainability [is] measured by the percentage of e-commerce sales to total sales” (1.Introduction, rows 117-118). This is arguable; strong arguments, documentation and/or reference/s should be provided in this respect.
4. The statement “E-commerce sustainability can measure [Reviewer’s question: or can be measured by?] the long-term e-commerce performance” of Thai MSMEs (Abstract, rows 27-28) is disputable. It is recommended to reformulate; in addition, strong arguments, documentation and/or reference/s should be provided in this respect.
5. SMEs value of Thai e-commerce (pictured in Figure 1) is different than MSMEs value of Thai e-commerce (rows 79-83); this contradiction should be addressed.
6. Second section (2.Literature review) deals with theories on technology adoption and e-commerce within TOE (technological, organizational, and environmental factors) framework. Then, a number of hypotheses related to e-commerce sustainability of Thai MSMEs are formulated. However, the logic/s and relationship/s between:
(i) {e-commerce, sustainability} in general;
(ii) {e-commerce, sustainability} of MSMEs;
(iii) {e-commerce, sustainability} of Thai MSMEs,
should be further developed, well documented and clearly explained.
7. The Tobit linear model is rather old (Tobin, 1958; Goldberger, 1964); therefore, the linearity of the model used should be completely justified; in addition: if the analytical model (formulae 1 and 2) might be an intuitive, reasonable measure of e-commerce, then relationship {e-commerce, sustainability} is still to be demonstrated / documented / referenced (see preceding #8).
8. A graphic representation (scheme) of the research model used would be of help. It is strongly recommended to present and represent the variables of this model.
9. The method used was the questionnaire-based survey; the structure of the research instrument (questionnaire) should be presented.
10. It is not clear why retail sector and F&B service industry MSMEs (per Abstract, row 17) were chosen to be investigated; Are they more apt for e-commerce? Which are their peculiarities? Are there any differences between retail sector and F&B service industry? Are they distinct or overlapping? The issue of sample selection should be addressed, and its particular features have to be discussed.
11. The issue of sample representativeness should be addressed.
12. It is recommended to elaborate / explain how the dichotomy (separation) set of factors versus set of barriers.
13. It is strongly recommended to author/s to highlight the original elements of their study – as compared to the state-of-the-art international literature.
14. It is suggested to split the last section (Conclusions) in Conclusion/s per se, Implication/s, Limitation/s and further research paths.
15. Suggestion: All acronyms (F&B, ETDA, etc.) to be explained at their first appearance in the proposed paper.
Round 2
Reviewer 4 Report
Most issues were addressed satisfactorily.
Good job!
Wish authors success and good luck in further research!
This manuscript is a resubmission of an earlier submission. The following is a list of the peer review reports and author responses from that submission.